# A Two-stage Cascaded Deep Neural Network with Multi-decoding Paths for Kidney Tumor Segmentation

Tian He[1], Zhen Zhang[1], Chenhao Pei[1], and Liqin Huang[1(✉)]

College of Physics and Information Engineering, Fuzhou University, Fuzhou, China
hlq@fzu.edu.cn

**Abstract.** Kidney cancer is aggressive cancer that accounts for a large proportion of adult malignancies. Computed tomography (CT) imaging is an effective tool for kidney cancer diagnosis. Automatic and accurate kidney and kidney tumor segmentation in CT scans is crucial for treatment and surgery planning. However, kidney tumors and cysts have various morphologies, with blurred edges and unpredictable positions. Therefore, precise segmentation of tumors and cysts faces a huge challenge. Consider these difficulties, we propose a cascaded deep neural network, which first accurately locate the kidney area through 2D U-Net, and then segment kidneys, kidney tumors, renal cysts through Multi-decoding Segmentation Network(MDS-Net) from the ROI of the kidney. We evaluated our method on the 2021 Kidney and Kidney Tumor Segmentation Challenge (KiTS21) dataset. The method dice score achieves 93.40%, 68.32%, 64.26% for kidney, kidney mass, and kidney tumors, respectively. The model of cascade network proposed in this paper has a promising application prospect in kidney cancer diagnosis.

**Keywords:** Kidney · Kidney tumor Segmentation · Cascaded deep neural network · Multi-decoding

## 1 Introduction

Renal cell carcinoma(RCC) is a malignant tumor formed by the malignant transformation of epithelial cells in different parts of the renal tubule [6]. Its incidence accounts for 80% to 90% of adult renal malignant tumors, and the prevalence of men is higher than that of women [5,1,4]. The incidence of kidney cancer is closely related to genetics, smoking, obesity, hypertension, and anti-hypertensive therapy, which is second only to prostate cancer and bladder cancers among tumors of the urinary system. Accurate segmentation of tumors from 3D CT remains a challenging task due to the unpredictable shape and location of tumors in the patient, as well as the confusion of textures and boundaries. [9,19].

The traditional method of manually segmenting tumors is not only time-consuming and laborious, but also has the problem of inconsistent results during segmentation by senior doctors, which leads to unsatisfactory results in clinical

applications [13,6,17]. Therefore, computer-assisted kidney tumor segmentation methods have attracted much attention.In recent years, deep learning has penetrated into various application fields, and its performance in many fields such as image detection, classification, and segmentation has exceeded the most advanced level [7]. Among current CNN-based methods,the popular U-Net [16] and 3D U-Net [3] architecture have exhibited promising results in medical image segmentation tasks [10], such as pancreas segmentation [14] , prostate segmentation [18] and brain segmentation [15]. Although 3D Fully Convolutional Network(FCN) [12] segmentation performance is higher than 2D FCN, it requires greater memory consumption. Zhang et al.[20] proposed a cascaded framework network for automatic segmentation of kidneys and tumors, which alleviates the problem of inaccurate segmentation caused by insufficient network depth due to excessive memory consumption. With extremely limited data,a cascaded 3D U-Net with a active learning function can improve training efficiency and reduce labeling work [8].

Recently, Li et al. [10] proposed a 3D U-Net based on memory efficiency and non-local context guidance, which captures the global context through a non-local context guidance mechanism and fully utilize long-distance dependence in the feature selection process.In the 3D U-Net, this method complements high-level semantic information with spatial information through a layer skip connection between the encoder and the decoder, and finally realizes the precise segmentation of the kidney and the tumor.

In this work, we develop a fully automatic cascaded segmentation network with multi-decoding paths. The kidney area is first located through 2D U-Net. The area is cropped according to the region of interest located in the first stage and input it into MDS-Net to accurately segment the kidney, renal tumor and renal cyst. General, the contributions of our work can be summarized in the following three aspects:

1. We develop a two-stage cascaded segmentation network with multi-decoding paths and evaluate it on the 2021 Kidney and Kidney Tumor Segmentation Challenge (KiTS21) dataset.
2. We propose a fusion module based on global context (GC)[2], which can realize the attention to channel and spatial context to achieve noise suppression and enhancement of useful information.
3. We present a regional constraint loss function, which is used to measure the constraint relationship of impassable regions.

## 2   Methods

Figure 1 shows the two-stage cascaded deep neural network for kidney tumor segmentation. First input the pre-processed image to locate the kidney through 2D U-Net to obtain an accurate kidney area, then use the kidney area as the bounding box of the original CT, cropping to get the input image, and train the MSD-Net to segment kideney, tumors and cysts.

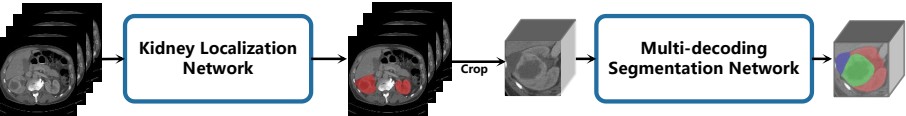

**Fig. 1.** The Kidney Localization Network (U-Net)

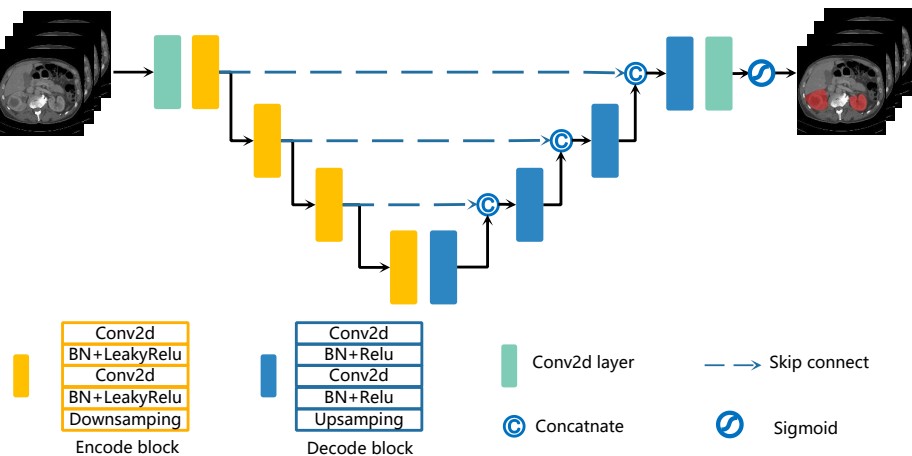

**Fig. 2.** The Kidney Localization Network (U-Net)

### 2.1 Kidney Localization Network

For the localization of the kidney, we trained a 2D U-Net for kidney segmentation and localization. As shown in Figure 2, the encoding path composed of four encoder blocks, and each block is composed of 2D convolution, Batchnorm, LeakyReLU and downsampling. On the decoding path, each decoding block is composed of convolution, Batchnorm, ReLU, and upsampling. After upsampling on the last layer, the image undergoes a 3×3 convolution. The input of the network is a 256×256 image, and the output is divided into the background and the kidney through a softmax function. The loss function used is Dice loss

$$\mathcal{L}oss_{KI} = 1 - DSC(L_{KI}, \hat{L}_{KI}), \tag{1}$$

where $DSC(A, B)$ calculates the Dice similarity coefficient of $A$ and $B$, $L_{KI}$ and $\hat{L}_{KI}$ are the corresponding gold standard and predicted label of whole regions including the kidneys, tumors, and cysts. For the segmentation results, we performed connected components analysis and selected the largest two connected component to locate the kidney.

### 2.2 Multi-decoding Segmentation Network

Multi-decoding segmentation network (MDS-Net) is designed to segment normal kidneys, kidney tumors, and kidney cysts. The Figure 3 shows the design

of MDS-Net, which consists of an encoding path, three decoding paths, and a fusion prediction branch. The image patch obtained by the first stage cropping

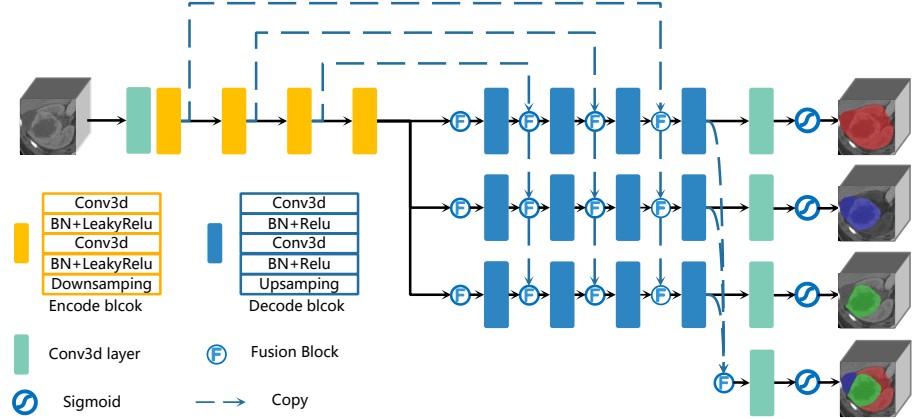

**Fig. 3.** Multi-decoding Segmentation Network

is input to the encoding path for feature extraction, and three segmentation results $(\hat{L}_{KI}, \hat{L}_{MA}, \hat{L}_{TU})$ are obtained by the three decoding paths. Fusion of the features obtained by the three decoding paths to obtain the final segmentation result $\hat{L}_{KTC}$.

Due to the imbalance of the segmentation classes, e.g. the cyst does not exist in any cases or only occupies a small area, which makes the network difficult to train. Therefore, we set the three regions of the target segmentation as $KI$ is the entire kidney region, including normal kidney, tumor and cyst, $MA$ is kidney masses that include tumors and cysts region, and $TU$ is the region of tumors only. By decomposing the original multi-label segmentation task into these three single-label segmentation tasks, the impact of category imbalance is reduced.

In the encoding path, feature extraction is performed by a convolutional layer and four encode blocks, and each encode block is composed of a 3D convolutional layer, Batchnorm, LeakyRelu, and downsampling. The future map obtained after downsampling is used as the input of the next module, and is also input into the decoding path through a skip connection. Each decoding branch is composed of a feature global context fusion block(see 2.3 for details), decoder, and a convolutional layer. The decode block is similar to the encode block, while the last layer is upsampling. The fusion block fuses and corrects the output feature maps from the previous layer and skip connection. The features obtained by the three decoding paths are output through the Sigmoid layer to obtain $(\hat{L}_{KI}, \hat{L}_{MA}, \hat{L}_{TU})$ separately, the corresponding loss functions are

$$\mathcal{L}oss_{KI} = 1 - DSC(L_{KI}, \hat{L}_{KI}), \tag{2}$$

$$\mathcal{L}oss_{MA} = 1 - DSC(L_{MA}, \hat{L}_{MA}), \tag{3}$$

$$\mathcal{L}oss_{TU} = 1 - DSC(L_{TU}, \hat{L}_{TU}), \tag{4}$$

where $(L_{KI}, L_{MA}, L_{TU})$ is the ground truth of region $(KI, MA, TU)$. Finally, the feature map output by each layer of decoding path is fused using the fusion module, and then the final segmentation result is predicted, and the loss function is

$$\mathcal{L}oss_{KTC} = 1 - DSC(L_{KTC}, \hat{L}_{KTC}) + Loss_{RC}, \tag{5}$$

$L_{KTC}$ is defined as the ground stand of the three categories kidneys, kidney tumors, kidney cysts. So the loss function of the entire network is

$$\mathcal{L}oss = Loss_{KTC} + Loss_{KI} + Loss_{MA} + Loss_{TU} + Loss_{RC}, \tag{6}$$

$Loss_{RC}$ is the regional constraint loss(see 2.4 for details).

### 2.3    Global Context Fusion Block

Inspired by [21,11], the GCFB is designed to fuse and calibrate feature maps to achieve noise suppression and enhancement of useful information. As shown in Figure 4, the Global Context (GC) block that combines Nonlcoal and Sequeze and Excitation (SE) block is used to calibrate the features, which can realize the attention to the channel and spatial context, and obtain the fused feature map through the convolutional layer and Relu.

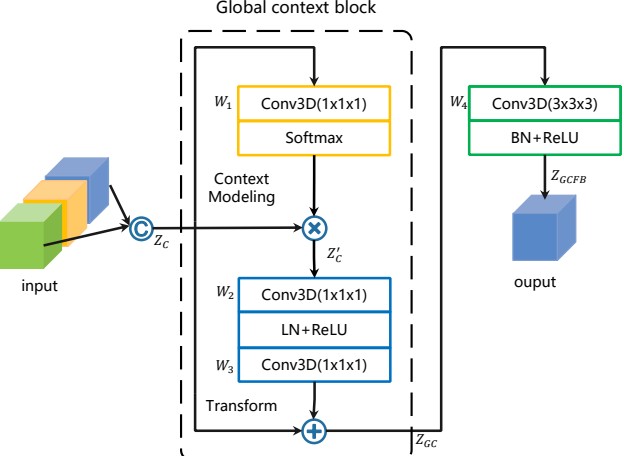

**Fig. 4.** Global Context Fusion Block

In GCFB, first concat the input on the channel to get $Z_C$. The GC blcok can be defined as:

$$Z_C^{'} = Z_C \otimes Softmax(W_1 Z_C), \tag{7}$$

$$Z_{GC} = Z_C + W_3 ReLU(LN(W_2 Z_C')), \tag{8}$$

$W_*$ is the parameters of the convolution layers. Therefore, the final ouput of GCFB is

$$Z_{GCFB} = ReLU(BN(W_4 Z_{GC})). \tag{9}$$

### 2.4   Regional Constraint Loss Function

As shown in Figure 5, there are regional relationships in different regions of the kidney, the region of $KI$ contains $MA$, $TU$ is in $MA$. In order to achieve

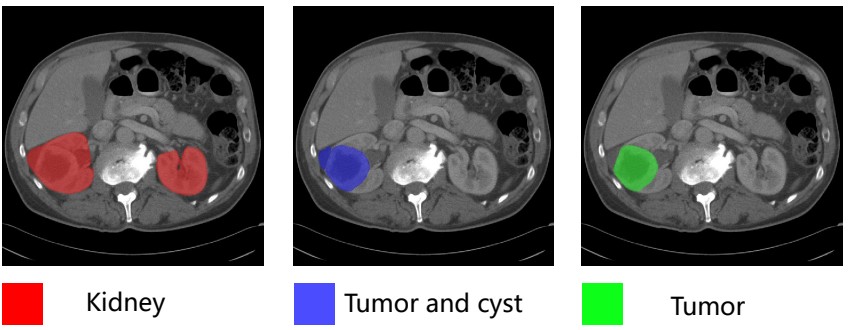

Kidney  Tumor and cyst  Tumor

**Fig. 5.** Regional constraint of kidney

the constraint of this relationship, the overlap degree of different regions[11] is calculated to measure whether the constraint relationship between the regions is satisfied. The regional constraint loss function is

$$\mathcal{L}oss_{RC} = 1 - \frac{1}{2}\left(\frac{\sum\limits_{x \in \Omega} \hat{L}_{KI}(x) \cdot \hat{L}_{MA}(x)}{\sum\limits_{x \in \Omega} \hat{L}_{MA}(x)} + \frac{\sum\limits_{x \in \Omega} \hat{L}_{MA}(x) \cdot \hat{L}_{TU}(x)}{\sum\limits_{x \in \Omega} \hat{L}_{TU}(x)}\right), \tag{10}$$

where $(\hat{L}_{KI}, \hat{L}_{MA}, \hat{L}_{TU})$ is predicted result from three decoding path from MDS-Net(see Figure 3), the $\Omega$ is the common spatial space.

## 3   Experimental Results

### 3.1   Dataset

The KiTS21 Challenge provides contrast-enhanced CT scans and annotation data from 300 patients who underwent partial or radical nephrectomy for suspected renal malignant tumors at M Health Fairview or Cleveland Clinic Medical Center between 2010 and 2020,which provides us with three annotation data,

and we finally chose the voxel-wise majority voting aggregations for training and validation. The size, shape and density of the tumor are various in different CT scans. Moreover, only a few images of existing cysts. The annotation work is completed by experienced experts, trainees and laid-off workers together, and the annotated data is used as the ground truth of the training.Since only the training set data was provided, we randomly divided the data of the 300 cases into 5 pieces, each with 60 cases. One of the 5 pieces was selected in turn as the verification set and the other 4 pieces as the training set.

### 3.2   Implementation Details

**Data processing:** Before training our cascade model, we first performed a crop slice operation to make all volume slices the same thickness to reduce GPU memory consumption and training time. For the first network, we input 2D axial slices, which are obtained by extracting slices from the original 3D CT along the z-axis and adjusting the size from $512 \times 512$ to $256 \times 256$. For the second network MDS-Net, according to the maximum rectangular frame range of the region of each kidney, the size of the block of the region of interest extracted is $128 \times 128 \times 128$. Then, we truncated the image intensity values of all images to the range of [-100,500] HU to remove the fat area around the kidney and remove irrelevant details.

**Implementation Details:** As an experimental environment, we choose Pytorch to implement our model and use NIVIDIA Tesla P100 16GB GPU for training.The input size of the first network is $256 \times 256$ with a batch size of 16, The input size of the second network is $128 \times 128 \times 128$ with batch size of 4. In our model, we set the epoch to 200, and the initial learning rate is $1 \times 10^{-2}$.

**Evaluation metrics:** We employ the DSC and the Surface Dice provided in the KiTS21 toolkit as the primary evaluation criteria for evaluating segmentation performance. For KiTS21, the following hierarchical classes are used to evaluate the DSC and the Surface Dice.

- Kidney and Masses ($KI$): Kidney + Tumor + Cyst,
- Kidney Mass ($MA$): Tumor + Cyst,
- Tumor ($TU$): Tumor only.

### 3.3   Results

To evaluate the effectiveness of our method, we compared our network with other state-of-the-art methods, including 2D U-Net and 3D U-Net. Furthermore, to explore the advantage of GCFB and $\mathcal{L}oss_{RC}$, we also compared our method with the method without GCFB and $\mathcal{L}oss_{RC}$. We perform visual and statistical comparisons under the same data set and data parameters. In addition, in order to explore the advantages of our method, we use the evaluation index Dice

**Table 1.** Dice score and Surface Dice of the proposed method and other baseline methods on the validation set.

| Method | Dice (%) | | | Surface Dice | | |
|---|---|---|---|---|---|---|
| | KI | MA | TU | KI | MA | TU |
| 2D U-Net | 90.78 | 43.61 | 44.86 | 83.30 | 28.21 | 29.51 |
| 3D U-Net | 91.59 | 59.29 | 57.71 | 83.76 | 40.55 | 39.95 |
| Ours (wo GCFB) | 92.55 | 63.23 | 58.63 | 83.84 | 42.11 | 39.11 |
| Ours (wo $\mathcal{L}oss_{RC}$) | 93.12 | 67.91 | 62.78 | 85.75 | **46.37** | 42.25 |
| Ours | **93.48** | **68.32** | **64.26** | **86.29** | 45.90 | **43.34** |

coefficient and the Surface Dice to verify our method on the KiTS 21 data set.We compared the results of our method with four different methods:

From Table 1, compared to 2D U-Net and 3D U-Net, our proposed methods performed better both in Dice and Surface Dice. It also that our proposed methods with GCFB and $\mathcal{L}oss_{RC}$ can achieve better results than our methods without GCFB or $\mathcal{L}oss_{RC}$. In addition, Figure 6 shows the visualization results of different methods. In Figure 6, our method is more effective than other methods in easier cases and challenging cases.

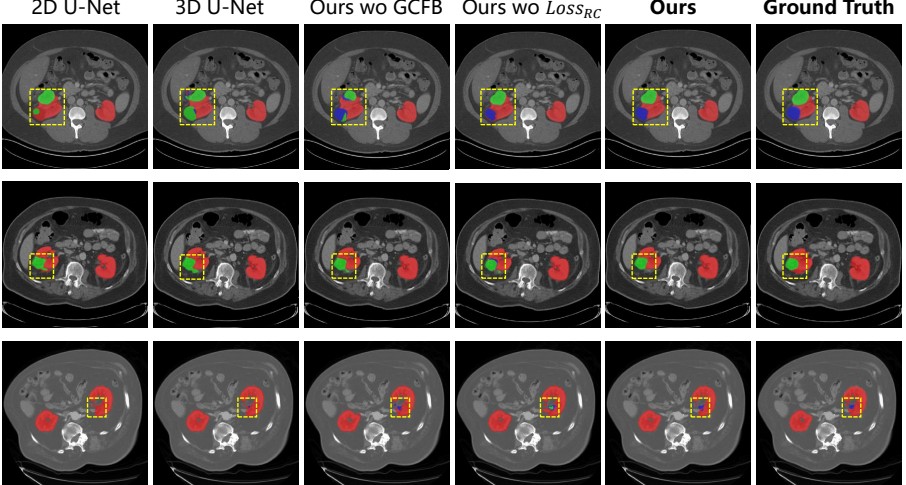

**Fig. 6.** Visualization results of different methods.The segmentation contained in the yellow dashed box is our concern.

## 4 Conclusion

In this work, we proposed a novel two-stage cascade and multi-decoding method for kidney segmentation. We utilized Unet to achieve the location and extraction

of the region of kidney, and then designed MSD-Net for the final segmentation. For MSD-Net, we developed a segmentation network with multiple decoders, combined the features of the three decoding paths with GCFB, and obtained the final segmentation result. Besides, we presented a regional constraint loss function to predict the segmentation result with more reality. It has been evaluated on the dataset from KITS 2021. Experimental results show that this method obtains good segmentation results on kidney tumors.

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
