# OpenReview forum: "A Two-stage Cascaded Deep Neural Network with Multi-decoding Paths for Kidney Tumor Segmentation"
_MICCAI.org/2021/Challenge/KiTS — Submitted to KiTS21 Challenge_

### Official Review · Reviewer_cNaS · 2021-08-30

**Rating:** 7

**Review:**

This paper does a great job giving a detailed background on the problem and giving an in-depth description of what was done and the results. A number of small grammatical errors are present but they do not detract from the reader's ability to understand the work. One crucial detail that is missing is how the authors handled the presence of multiple annotations per case. Did they use the majority voting aggregations for training and validation? If so, they should state as much within the paper. It would also be nice if they could include the official results once they are known.

---

### Official Review · Reviewer_zFQm · 2021-08-30

**Rating:** 8

**Review:**

### Overall

- Excellent level of detail. Nice work.

### Introduction

- Great job writing a detailed background for this problem. My only comment is about the first statement in the abstract "Kidney cancer is an aggressive cancer that accounts for a large proportion of adult renal malignancies" - of course kidney cancer accounts for most renal malignancies, perhaps you just mean "... of adult malignancies"?

### Methods

- Very nice figure
- Does your 2d network run on the axial slices only?
- I believe it's more typical to say "Batchnorm" rather than "Batchnormal"

### Results

- Very nice job with this section

### Discussion and Conclusion

- I believe it would be more correct to say this method achieves "good segmentation results" rather than "well segmentation results"

---

### Decision · Program_Chairs · 2021-08-30

**Decision:**

Minor Revisions

**Comment:**

Please address the reviewer comments and resubmit